# Impact of Short-Term Creatine Supplementation on Muscular Performance among Breast Cancer Survivors

**DOI:** 10.3390/nu16070979

**Published:** 2024-03-27

**Authors:** Emily J. Parsowith, Matt S. Stock, Olivia Kocuba, Alec Schumpp, Kylah Jackson, Alexander M. Brooks, Alena Larson, Madison Dixon, Ciaran M. Fairman

**Affiliations:** 1Cognition, Neuroplasticity, and Sarcopenia (CNS) Lab, Institute of Exercise Physiology and Rehabilitation Science, University of Central Florida, Orlando, FL 32816, USA; emily.parsowith@ucf.edu (E.J.P.); matt.stock@ucf.edu (M.S.S.); 2Exercise Oncology Lab, University of South Carolina, Columbia, SC 29208, USA

**Keywords:** creatine, breast cancer, oncology, muscular performance, strength, supplementation, cancer treatment

## Abstract

Breast cancer (BC) is one of the most common cancers in the United States. Advances in detection and treatment have resulted in an increased survival rate, meaning an increasing population experiencing declines in muscle mass and strength. Creatine supplementation has consistently demonstrated improvements in strength and muscle performance in older adults, though these findings have not been extended to cancer populations. PURPOSE: The purpose of this study was to investigate the effects of short-term creatine supplementation on muscular performance in BC survivors. METHODS: Using a double-blind, placebo-controlled, randomized design, 19 female BC survivors (mean ± SD age = 57.63 ± 10.77 years) were assigned to creatine (SUPP) (*n* = 9) or dextrose placebo (PLA) (*n* = 10) groups. The participants completed two familiarization sessions, then two test sessions, each separated by 7 days, where the participants supplemented with 5 g of SUPP or PLA 4 times/day between sessions. The testing sessions included sit-to-stand power, isometric/isokinetic peak torque, and upper/lower body strength via 10 repetition maximum (10RM) tests. The interaction between supplement (SUPP vs. PLA) and time (Pre vs. Post) was examined using a group × time ANOVA and effect sizes. RESULTS: No significant effects were observed for sit-to-stand power (*p* = 0.471; η_p_^2^ = 0.031), peak torque at 60°/second (*p* = 0.533; η_p_^2^ = 0.023), peak torque at 120°/second (*p* = 0.944; η_p_^2^ < 0.001), isometric peak torque (*p* = 0.905; η_p_^2^ < 0.001), 10RM chest press (*p* = 0.407; η_p_^2^ = 0.041), and 10RM leg extension (*p* = 0.932; η_p_^2^ < 0.001). However, a large effect size for time occurred for the 10RM chest press (η_p_^2^ = 0.531) and leg extension (η_p_^2^ = 0.422). CONCLUSION: Seven days of creatine supplementation does not influence muscular performance among BC survivors.

## 1. Introduction

In 2018, over 2 million new cases of breast cancer were diagnosed worldwide, and over 600,000 deaths were recorded. Advances in the detection and treatment of breast cancer have resulted in an increased survival rate, with current estimates of the 5-year survival for all stages at approximately 90% [1]. Unfortunately, this has resulted in an increased number of individuals experiencing declines in physiological wellbeing as a result of cancer treatments [2,3]. Individuals with breast cancer often experience reductions in muscle mass, strength, and physical function with cancer treatments [4]. In addition to physiological decline, individuals with breast cancer often experience a decline in cognitive function that persists years after treatment cessation [5]. This exponential decline in health in individuals increases the risk of disability, loss of independence, and mortality [6]. Consequently, there is a critical need to develop therapeutic interventions that offset the decline in physical function experienced following breast cancer treatments.

Creatine is a naturally occurring compound found in skeletal muscle that is essential for energy production during rest and exercise [7]. While creatine can be consumed through a diet high in meat and fish [8,9], supplementation with creatine monohydrate is typically required to fully saturate creatine stores [10]. A “loading phase”, defined as supplementing with creatine monohydrate for 5–7 days at 20–25 g per day [11], has been shown to rapidly increase intramuscular creatine stores [12,13], similar to a maintenance dose over 28 days [11]. Increasing intramuscular phosphocreatine stores contributes to an increase in ATP production and a resulting increase in exercise capacity, particularly during high-intensity exercise [14,15,16]. Creatine monohydrate is the most widely studied supplement to date, showing improvements in muscle strength, mass, and power in a variety of healthy, clinical, and older adult populations when consumed chronically [7,17,18]. Moreover, high-dose creatine supplementation has been shown to improve many cognitive domains, such as short-term memory and intelligence/reasoning, which may attenuate the exponential decline in cognitive function that many breast cancer patients experience [19]. Furthermore, experimental evidence has shown that short-term creatine monohydrate supplementation can also result in meaningful improvements in muscle creatine levels, muscle mass, and muscle strength [6,7,17].

While extensive research has been conducted on the short-term effects of creatine monohydrate supplementation in healthy young adults [14], along with older adult populations [20], research investigating the effectiveness of creatine monohydrate supplementation in oncology is in its infancy [21,22]. Individuals with cancer typically experience a blunted response to resistance training relative to age-matched individuals without a cancer diagnosis; therefore, it is unclear whether creatine monohydrate supplementation can augment these adaptations [23]. Given the decline in muscle performance that occurs with breast cancer treatments and the overwhelming evidence in support of creatine monohydrate supplementation on these outcomes in relevant populations, there is a clear and strong rationale for the investigation of creatine monohydrate supplementation on muscle performance in individuals with breast cancer. Thus, the primary aim of our study was to investigate the impact of short-term (~7 days) creatine monohydrate supplementation (20 g) on muscular performance in individuals previously treated for breast cancer. It was hypothesized that short-term supplementation would yield improvements in lower-body muscular power and strength.

## 2. Methods

### 2.1. Experimental Approach to the Problem

This study utilized a randomized, double-blind, placebo-controlled design to examine the effects of short-term creatine supplementation on muscular performance in individuals with breast cancer. An overview of the study design can be found in Figure 1. The data for this study were collected in the Exercise Oncology Laboratory at the University of South Carolina. Each participant underwent two familiarization sessions, separated by at least 72 h, prior to any experimental condition to familiarize them with the testing protocol and minimize practice effects (Figure 1). The participants performed a battery of tests including sit-to-stand power, balance, chair stand, gait speed, timed up-and-go, and 10 repetition maximum (10RM) for the chest press and leg extension. Knee extensor isometric and concentric isokinetic strength tests were also examined using an isokinetic dynamometer. The time of each testing battery was standardized within, but not between, the participants. Detailed descriptions of each assessment are provided below and are described in the order in which the tests were performed. Each participant was randomly allocated in a ratio of 1:1 to either the treatment (creatine) or placebo groups at the end of the first experimental session (T1). Randomization was conducted by a blinded member of the study team, with no contact with the study participants, using an online randomization generator. The participants and all other study staff were blinded to the allocation of groups.

### 2.2. Participants

A convenience sample of *n* = 19 females in Columbia, South Carolina (USA) previously treated for breast cancer was recruited to participate in this study using posted flyers at the local oncology facilities and community cancer-related events. Interested participants completed an online screening form regarding their health status, current intake of nutritional supplements, and previous exercise history. Participants were excluded from the study if they were actively receiving breast cancer treatment, the location posed a problem, or they were currently supplementing with creatine. Additionally, participants were excluded if their health status was deemed unsafe for exercise and supplementation, posing a risk to the individual (e.g., previous injury, cardiovascular condition, or bone metastases). Interested and eligible individuals gave informed consent prior to the start of any study activities. This study was approved by the University of South Carolina’s Institutional Review Board (Pro00119366).

### 2.3. Sit-to-Stand (STS) Power

This study utilized a chair and linear transducer (Tendo Weightlifting Analyzer, Trencin, Slovak Republic) to measure STS power [24]. The participants were asked to perform several practice attempts prior to testing. To begin the STS power measurement, a belt attached to the Kevlar string from the unit was secured around the participant’s waist, and the participant was positioned so that the string was perpendicular to the floor when standing. Then, the participant was instructed to sit in the middle of the chair with their arms folded across their chest, and markings were placed at their feet position to allow for repeated testing. The participant was instructed to stand up as quickly as possible before sitting back down. The power was calculated by the transducer using the vertical velocity (m/s) and mass moved (kg). The test was performed 3 times with a short rest in between, and the highest of the 3 peak power values was recorded [25].

### 2.4. 10 Repetition Maximum Strength

Upper body and lower body strength were assessed using 10 repetition maximum (10RM) tests with the chest press and leg press, respectively. The participants were asked to perform a full-body warm-up followed by an exercise-specific warm-up on both the chest press and leg extension machines. The exercise-specific warm-up sets included 4–6 repetitions at increasing loads (moving towards an anticipated 10RM load), each separated by 90–180 s. The participants were then instructed to complete a maximal attempt of 10 repetitions (i.e., the maximum weight an individual could lift for 10 repetitions through a full range of motion). The 10RM was attempted multiple times at increasing weights, with adequate rest, until the participant no longer used the correct technique [21]. Every effort was made to ensure a 10RM was achieved with as few attempts as possible to minimize the impact of fatigue on the results. The highest weight lifted at proper technique was recorded as the participant’s 10RM.

### 2.5. Maximal Voluntary Isometric and Concentric Isokinetic Knee Contractions

The maximal isometric and concentric isokinetic peak torque of the knee extensors was quantified using a Biodex System 3 isokinetic dynamometer (Biodex Medical Systems, Shirley, NY, USA). Testing was performed unilaterally on the dominant side (based on kicking preference). The participants were seated with their dominant knee, shoulders, hips, and dominant ankle stabilized with restraining straps. The input axis of the dynamometer was aligned with the axis rotation of the knee. Each participant’s chair settings were recorded during the initial familiarization visit and utilized for subsequent testing visits.

The participants first performed maximal concentric isokinetic contractions [26]. The participants started with their knees at 90°, with an end range of motion of 160°. Two angular velocities were used to test peak torque: 60°/s and 120°/s. After three submaximal efforts, the participants performed five flexion/extension repetitions at each angular velocity, with the goal of producing maximum torque. For each velocity, the peak torque value achieved was recorded (Nm). These are common tests of maximal knee extension strength and are regularly used with older adults and clinical populations [27,28,29,30].

Following the maximal concentric isokinetic contractions, the participants performed maximal voluntary isometric contractions (MVICs) with the knee flexed at 120° (60° below the horizontal plane) [31]. Prior to MVIC testing, both the away/toward limits and limb weight were re-calculated. Each participant then performed three five-second MVICs, each separated by 60 s of rest. The highest peak torque generated across the three attempts was recorded as the MVIC (Nm).

### 2.6. Supplementation

The participants randomized into either the creatine (SUPP) or placebo (PLA) received began the supplementation protocol immediately after the first testing visit. A blinded member of the study team not involved with the assessments was responsible for group allocation, and opaque envelopes containing the supplement were coded so that neither the investigators nor the participants were aware of the contents. Group assignment was not identified until after statistical analysis had been completed for all the outcomes. Between the two testing sessions, the SUPP group consumed 20 g of creatine monohydrate powder (Optimum Nutrition, Middlesbrough, UK), whereas the PLA group consumed 20 g of dextrose powder each day for 7 days. The rationale for the dose of 20 g/day was based on prior research demonstrating that this dose was sufficient to increase intramuscular creatine stores after 5–7 days [32]. Additionally, this dose has been used in numerous prior studies examining the short-term impact of creatine supplementation on muscle performance [12,13,33,34]. 

The participants were asked to consume 5 g of the supplement with ~250 mL of zero/low-calorie fruit juice 4 times per day to mask the flavor and solubility. We recommended taking smaller doses of creatine rather than the full loading dose at one time due to the decreased risk of gastrointestinal issues associated with a 5 g dose multiple times per day [35]. Each participant received 28 individually packaged 5 g doses to ensure accurate supplementation. A text message was sent to each participant at 8:00 AM, 12:00 PM, 4:00 PM, and 8:00 PM as a reminder to take each of the 4 doses of the supplement. Each participant was instructed to consume their normal diet throughout the duration of the study. The participants then returned to the laboratory with the envelope with empty supplement bags for a second round of testing (T2). Compliance with supplementation was conducted by verifying the ratio of empty packets returned to the investigators at T2 relative to the number of bags provided.

### 2.7. Statistical Analyses

The descriptive statistics are expressed as the mean ± standard deviation. To examine pre–post changes for each of the outcomes (body mass (kg), sit-to-stand power (W), MVIC and concentric isokinetic peak torque of the knee extensors, 10RM chest press (kg), and 10RM leg extension (kg)) between the groups and the time, a two-way (group [SUPP, PLA] × time [pre, post]) mixed factorial analysis of variance (ANOVA) was conducted. Group (SUPP or PLA) served as a between-subjects factor, whereas time (pre, post) served as a within-subjects factor.

In the event of a two-way interaction or main effects for a group or time, Bonferroni-corrected pairwise comparisons were examined. An alpha level of 0.05 was used to determine statistically significant differences. In addition to null-hypothesis significance tests, effect sizes via Cohen’s *d* statistics (for pairwise comparisons) and partial eta squared (η_p_^2^ for ANOVAs) were computed and evaluated. Small, medium, and large Cohen’s *d* values corresponded to 0.20, 0.50, and 0.80, respectively, whereas small, medium, and large partial eta squared values corresponded to 0.01, 0.06, and 0.14, respectively [36]. JASP software (version 0.16, The JASP Team, 2019) was used for all the statistical analyses [37].

## 3. Results

### 3.1. Participants

The participants who completed the study (*n* = 19) were all female, with a mean age of 57.63 ± 10.48 years and a mean BMI of 29.97 ± 5.79 kg/m^2^ (calculated using BMI = body mass (kg)/height^2^ (m^2^)) [38]. Their ethnicities were as follows: 79% White, 5% Asian, 11% Black, and 5% Hispanic. In total, 11 participants were stage I (58%), 6 were stage II (32%), 2 were stage III (10%), and the average time since diagnosis was 35.37 ± 16.93. Select participant characteristics are shown in Table 1. Both groups were considered homogenous at baseline. Pre-post values and mean differences within groups are expressed in Table 2.

### 3.2. Body Mass

The results from the two-way mixed factorial ANOVA indicated that there was no significant time × group interaction (*F* = 2.147, *p* = 0.161, η_p_^2^ = 0.122) as well as no main effect for the group (*F* = 0.111, *p* = 0.743, ή^2^ = 0.007) or time (*F* = 0.860, *p* = 0.367, η_p_^2^ = 0.048).

### 3.3. Sit-to-Stand Power

The results from the two-way mixed factorial ANOVA indicated that there was no significant time × group interaction (*F* = 0.544, *p* = 0.471, η_p_^2^ = 0.031) as well as no main effect for the group (*F* = 0.017, *p* = 0.898, ή^2^ < 0.001) or time (*F* = 2.318, *p* = 0.146, η_p_^2^ = 0.120).

### 3.4. MVIC Peak Torque

The results from the two-way mixed factorial ANOVA indicated that there was no significant time × group interaction (*F* = 0.015, *p* = 0.905, η_p_^2^ < 0.001) as well as no main effect for the group (*F* = 02.304, *p* = 0.147, ή^2^ = 0.119) or time (*F* = 1.861, *p* = 0.190, η_p_^2^ = 0.099).

### 3.5. 60°/Second Concentric Isokinetic Peak Torque

The results from the two-way mixed factorial ANOVA indicated that there was no significant time × group interaction (*F* = 0.404, *p* = 0.533, η_p_^2^ = 0.023) as well as no main effect for the group (*F* = 0.513, *p* = 0.484, ή^2^ = 0.029) or time (*F* = 2.404, *p* = 0.139, η_p_^2^ = 0.124).

### 3.6. 120°/Second Concentric Isokinetic Peak Torque

The results from the two-way mixed factorial ANOVA indicated that there was no significant time × group interaction (*F* < 0.001, *p* = 0.994, η_p_^2^ < 0.001) as well as no main effect for the group (*F* = 0.040, *p* = 0.843, ή^2^ = 0.002) or time (*F* = 0.038, *p* = 0.848, η_p_^2^ = 0.002).

### 3.7. 10RM Chest Press

The results from the two-way mixed factorial ANOVA indicated that there was no significant time × group interaction (*F* = 0.723, *p* = 0.407, η_p_^2^ = 0.041) as well as no main effect for the group (*F* = 0.177, *p* = 0.679, η_p_^2^ = 0.010). There was, however, a significant main effect for time (*F* = 19.218, *p* < 0.001, η_p_^2^ = 0.531). The results from the Bonferroni corrected pairwise comparison indicated that, when collapsed across the group, the 10RM chest press strength increased from (displayed as mean ± standard error) 19.8 ± 1.1 to 21.5 ± 1.1 kg.

### 3.8. 10RM Leg Extension

The results from the two-way mixed factorial ANOVA indicated that there was no significant time × group interaction (*F* = 0.007, *p* = 0.932, η_p_^2^ < 0.001) as well as no main effect for the group (*F* = 0.072, *p* = 0.791, η_p_^2^ = 0.004). There was, however, a significant main effect for time (*F* = 12.430, *p* = 0.003, η_p_^2^ = 0.422). The results from the Bonferroni corrected pairwise comparison indicated that, when collapsed across the group, the 10RM leg extension strength increased from (displayed as mean ± standard error) 23.8 ± 1.6 to 25.4 ± 1.6 kg.

## 4. Discussion

Though previous studies have demonstrated that creatine supplementation improves muscular strength, mass, and power in healthy, clinical, and older adult populations [7,17,18], a considerable gap exists in our understanding of the effectiveness of creatine supplementation in cancer populations. This study was designed to examine the impact of a 7-day short-term creatine monohydrate supplementation on muscular performance in individuals previously treated for breast cancer. The primary findings from our study were that short-term creatine supplementation had no effect on muscular performance as measured by sit-to-stand power, MVIC peak torque, isokinetic peak torque, 10RM chest press and leg extension and displayed no significant changes in both the SUPP and PLA groups. Specifically, there were no group, time, or group × time interaction effects for sit-to-stand power, MVIC peak torque, or isokinetic peak torque. There was a main effect for time for 10RM chest press and leg extension, with no differences between the groups.

Previous research into the effects of short-term creatine supplementation has been relatively mixed, with some studies showing improvements in muscular performance [39,40,41,42,43,44,45,46,47] and others showing no effect [48,49,50]. The ergogenic effects of creatine are hypothesized to be most relevant in activities that are high intensity and of short duration, where the ATP-PC system is the predominant energy system and supplementation allows for quicker regeneration of ATP to enhance performance. It has been previously demonstrated that high-dose supplementation (i.e., 20 g/d for ~4 days) can increase intramuscular phosphocreatine stores by ~6%. It has been observed previously that there may be non-responders to high-dose supplementation, with suggestions that ~30% of individuals do not respond to creatine loading protocols with resultant increases in PCr [51]. Responders to Cr supplementation typically have lower levels of initial PCr, a higher percentage of type II fibers, and overall fat-free mass, which can be negatively impacted by cancer treatments [52,53,54]. We did not measure intramuscular creatine stores, so it is impossible for us to determine if/what change in intramuscular PCr was observed in our participants. Nevertheless, a potential explanation for the lack of observed change in our outcomes could be the variability in the responses to creatine supplementation.

Another potential explanation for the lack of differences observed in our study could be the history of cancer. It has been well established that individuals with cancer exhibit an “accelerated aging” phenotype, whereby it is expected to observe a ~10-year biological aging in individuals who receive chemotherapy [55,56,57,58]. Moreover, it has been observed that individuals treated with chemotherapy experience a blunted response to exercise training [59,60]. Potential mechanisms for this have been proposed, including high-grade inflammation, mitochondrial dysfunction, oxidative stress, and alterations in molecular pathways of muscle remodeling [61,62,63]. Without assessing any of these directly in our study, it is unclear whether or not these may have impacted the effectiveness of the supplementation in our trial. Nevertheless, the fact that the majority of individuals in this study (79%) previously received chemotherapy is worth noting. Future research should look to fully explore the impact of chemotherapy treatment on the mechanisms involved in the ergogenic effect of creatine supplementation to further elucidate this effect.

There was a main effect of time for 10-RM testing in our study, but no group × time effects were observed. There were no effects detected for any of the other outcomes. Importantly, the acute ergogenic effect of creatine supplementation is suggested to be a result of saturated intramuscular creatine stores (from supplementation), allowing for quicker resynthesis of ATP during short-duration, high-intensity activities [64]. However, this hypothesis is reliant on the premise that ATP stores need to be sufficiently depleted from the activity, to where quicker resynthesis would actually be necessary to enhance performance. Most of the assessments selected in our battery took anywhere from 2–8 s to complete and were separated by 30–180 s of rest between repetitions and individual assessments. Consequently, it is plausible that a combination of the assessments selected, along with the rest taken between the test attempts, and each test, resulted in ATP not being depleted sufficiently, to where quicker replenishment of stores (because of increased PCr from creatine supplementation) was not necessary for performance improvements. Essentially, it is possible that the loading/intensity of the test battery was not high enough and the rest periods were sufficiently high such that ATP was able to be replenished sufficiently from rest alone, negating the impact of creatine supplementation.

As an extension of the above, it is well understood that the ergogenic effects of creatine supplementation on muscular performance are most potent when combined with resistance training and consumed chronically [64,65]. For example, the results of a recent meta-analysis indicated that creatine supplementation resulted in great increases in lean tissue mass, leg press strength, and chest press strength during resistance training compared to resistance training alone in older adults [66]. Increased PCr from supplementation can allow for quicker replenishment of ATP during short-duration high-intensity resistance training, with repeated bouts of effort (i.e., sets) separated by minimal rest. Future studies may look to explore the impact of short-term creatine supplementation on muscular performance during resistance exercise bouts that more closely mirror that of what would be performed during a typical resistance exercise session (i.e., multiple exercises, sets, and reps, with minimal rest). Furthermore, whilst the findings from our study indicated no ergogenic effect of short-term creatine supplementation on neuromuscular performance, the rapidly growing body of evidence supporting improvements in muscle mass and strength from combined creatine supplementation and resistance training compared to resistance training alone in older adults and other clinical populations, warrants continued investigation of its effectiveness in oncology settings.

This study of short-term creatine supplementation in individuals previously treated for breast cancer had several strengths worthy of note. The inclusion of two familiarization sessions prior to the first testing session helped to reduce the practice effect, allowing for a better understanding of the true effects of creatine supplementation. In addition, the double-blind study design prevented investigator bias. Our study is not without limitations. Firstly, we recruited a convenience sample of *n* = 19 individuals, primarily due to time and resource constraints. Nevertheless, the small sample size may have limited our ability to adequately detect changes in our outcomes. Furthermore, our sample comprised individuals who were previously treated for breast cancer, stages I-III. Consequently, these findings may not extend to the metastatic breast cancer setting nor to other tumor types. Additionally, we utilized a relatively short period (7 days) of supplementation. This dose/time period was chosen based on extensive evidence demonstrating that total creatine and phosphocreatine stores can be increased with ~20 g of creatine for 5–7 days [32]. Moreover, this study design was based on similar protocols from studies examining short-term creatine supplementation on muscular performance [12,13,33,34]. Nevertheless, it is possible that the timeframe used in this study was insufficient to detect an ergogenic effect. Consequently, future studies may look to explore longer timeframes of supplementation. Lastly, the absence of direct assessment of PCr precludes our ability to confirm the efficacy of the creatine loading protocol on intramuscular PCr stores.

## 5. Conclusions

In conclusion, the findings from our study demonstrated that short-term creatine supplementation did not improve muscular performance in women who were previously treated for breast cancer. Potential explanations for this are potential variations in the participants’ responses to creatine loading, the assessment battery selected, and/or the amount of recovery observed during/between the tests. We hope that these findings offer support to researchers looking to expand this research into the investigation of creatine supplementation in cancer populations.

## Figures and Tables

**Figure 1 nutrients-16-00979-f001:**
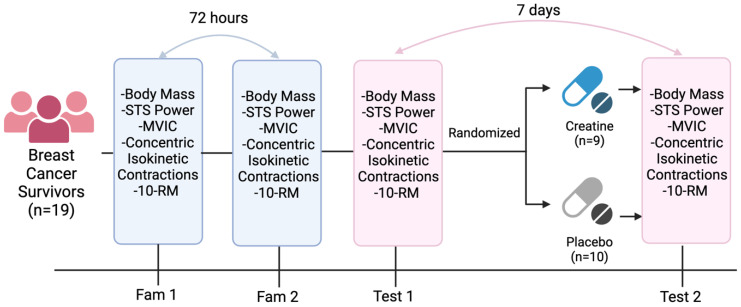
Timeline of Test Battery. Created using Biorender.com.

**Table 1 nutrients-16-00979-t001:** Participant Demographics.

Total *n*, %	All		Creatine (SUPP)		Placebo (PLA)	
	19	% or (SD)	9	% or (SD)	% or (SD)	100%
Age (yrs), mean (SD)	57.63	(10.48)	58.11	(11.22)	57.20	(9.74)
Body mass (kg), mean (SD)	76.04	(15.31)	77.20	(13.10)	74.99	(16.99)
Height (cm), mean (SD)	162.42	(4.82)	164.11	(5.80)	160.91	(3.01)
BMI (kg/m^2^), mean (SD)	29.97	(5.79)	30.91	(5.97)	28.09	(4.90)
Smoking status, *n*%						
No smoking since year before diagnosis	13	68%	5	56%	8	80%
Ex-smoker	6	32%	4	44%	2	20%
Cancer stage, *n* %						
I	11	58%	5	56%	6	60%
II	6	32%	3	33%	3	30%
III	2	10%	1	11%	1	10%
Cancer treatment, *n* %						
Surgery	19	100%	9	100%	10	100%
Chemotherapy	15	79%	7	78%	8	80%
Immunotherapy	2	21%	1	11%	1	10%
Hormone therapy	6	32%	3	33%	3	30%
Radiotherapy	16	84%	8	89%	8	80%
Osteoporosis or osteoarthritis, *n*%						
Yes	7	37%	2	11%	1	10%

**Table 2 nutrients-16-00979-t002:** Measured Outcomes (mean ± standard deviation).

Outcome	SUPP			PLA			*p* Value *
	Pre	Post	Mean Difference	Pre	Post	Mean Difference	
Sit-to-stand power (W)	1500.78 ± 635.11	1566.00 ± 636.03	65.22	1475.70 ± 614.59	1663.40± 642.76	187.70	0.47
60° concentric isokinetic peak torque (Nm)	68.38 ± 16.07	71.06 ± 16.47	2.68	61.20 ± 19.79	67.60 ± 16.64	6.40	0.53
120° concentric isokinetic peak torque (Nm)	52.48 ± 13.51	52.86 ± 10.64	0.38	51.28 ± 18.07	51.69 ± 10.17	0.41	0.99
MVIC peak torque (Nm)	85.53 ± 22.39	90.91 ± 30.19	5.38	69.78 ± 21.86	74.28 ± 23.19	4.50	0.91
10RM chest press (kg)	19.15 ± 3.23	21.16 ± 3.40	2.01	20.41 ± 5.66	21.77 ± 6.25	1.36	0.41
10RM leg extension (kg)	23.43 ± 4.94	24.94 ± 6.00	1.51	24.26 ± 8.49	25.85 ± 8.01	1.59	0.93

* *p* value for group × time interaction effects.

## Data Availability

The data presented in this study are available upon request from the corresponding author due to ethical reasons.

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
