# Peer review of "Impact of Short-Term Creatine Supplementation on Muscular Performance among Breast Cancer Survivors"

_nutrients, 2024, doi:10.3390/nu16070979_

Round 1
Reviewer 1 Report
Comments and Suggestions for Authors
Congratulations for the study. There is a large issue that I would like the authors to change.
Although the main idea is worthy and relevant, the whole paper is about creatine. there is no explanation (other than the treatment promotes muscle loss) of why should a cancer survivor use creatine.
Maybe the authors should describe better the participants, show a weight variation timeline to assure they could benefit from creatine.
If there is no connection with the cancer condition, no rationale of what is different or what could be different in the body because of cancer or its treatment that would justify the creatine supplementation, the authors could be using women that didn't have cancer.
Lastly, the study didn't show any differences. there is some explanation, but none of them connected, even remotely, to the cancer, or its treatment.
Does the lack of effects could be related to the cancer treatment??
This connection needs to be established.
Reviewer 2 Report
Comments and Suggestions for Authors
Congratulations on the work developed so far. Conducting research for breast cancer survivors is essential for advancing knowledge, improving care, and enhancing the lives of those affected by the disease.
In general, it is a well-conducted and presented study.
My only suggestion: Please explain and add some lines in the Discussion and Limitations part on why you choose to deliver creatine supplementation only for this short period (7 days). Would you expect different results during a more extended period, e.g., 14 days?
Also, provide some text in the Introduction part (paragraph 3) explaining the possible underlying mechanisms of creatine supplementation in the context of cancer survivorship, apart from the changes in muscle performance.
Reviewer 3 Report
Comments and Suggestions for Authors
Congratulations. The subject of this study is highly relevant. However, the presentation of the manuscript must be improved. Please, find in the attached file my comments and criticisms.

Reviewer 4 Report
Comments and Suggestions for Authors
-
1.The duration of creatine supplementation appears to be relatively short, spanning only 7 days. It raises concerns about whether this timeframe is sufficient for the participants to accumulate an optimal level of creatine reserves within their bodies to support improvements in muscle function.
-
2.There is uncertainty regarding the remarkably low body weight of the participants. It is unclear whether this discrepancy is due to an issue with the mobile device or if it accurately reflects the characteristics of the subjects. Further clarification or verification is necessary.
-
3.The rationale behind the dosage of 5g of creatine supplementation warrants discussion. Is this dosage scientifically justified, and should it be adjusted based on the participants' body weight, possibly adhering to a per kilogram basis? This aspect requires elaboration to ensure the appropriateness and efficacy of the supplementation protocol.
ok
Round 2
Reviewer 1 Report
Comments and Suggestions for Authors
Congratulations again! The observations I made were taken into consideration and answered when possible. The responses were satisfatory and the paper is now improved enough to be published.
Author Response
We would like to thank the reviewer for taking the time to greatly improve our manuscript. Each and every comment undoubtedly strengthened our research.
Reviewer 3 Report
Comments and Suggestions for Authors
Congratulations. The manuscript was improved.
Author Response
We would like to thank the reviewer for taking the time to review our manuscript. All of your comments were greatly appreciated and undoubtedly improved our paper.
Reviewer 4 Report
Comments and Suggestions for Authors
The innovation presented in this article is notably limited, as it fails to introduce significant new knowledge to the readership. Comparable studies abound, yet the research design employed here is overly simplistic and lacks depth, neglecting to elucidate the relevant underlying mechanisms. Consequently, we maintain our recommendation to reject the publication of this article.
Comments on the Quality of English Languagegood
Author Response
We would like to thank the reviewer for taking the time to review this manuscript. All comments and suggestions from Round 1 were addressed, improving the overall content and quality of the manuscript. The initial review showed no inclination that the paper should be rejected for publication. The authors are open to any further comments in order to prepare this manuscript for publication.